# Current Progress in Magnetic Resonance-Guided Focused Ultrasound to Facilitate Drug Delivery across the Blood-Brain Barrier

**DOI:** 10.3390/pharmaceutics16060719

**Published:** 2024-05-27

**Authors:** Ying Meng, Lorraine V. Kalia, Suneil K. Kalia, Clement Hamani, Yuexi Huang, Kullervo Hynynen, Nir Lipsman, Benjamin Davidson

**Affiliations:** 1Division of Neurosurgery, Department of Surgery, University of Toronto, Toronto, ON M4N 3M5, Canada; 2Harquail Centre for Neuromodulation, Hurvitz Brain Sciences Program, Sunnybrook Research Institute, Toronto, ON M4N 3M5, Canada; 3Sunnybrook Research Institute, Toronto, ON M4N 3M5, Canada; 4Krembil Research Institute, University Health Network, Toronto, ON M5T 0S8, Canada; 5Division of Neurology, Department of Medicine, University of Toronto, Toronto, ON M5S 1A1, Canada; 6Center for Advancing Neurotechnological Innovation to Application (CRANIA), University Health Network, Toronto, ON M5T 1M8, Canada; 7KITE Research Institute, University Health Network, Toronto, ON M5G 2A2, Canada

**Keywords:** focused ultrasound, MRgFUS, blood brain barrier, Parkinson’s disease, Alzheimer’s disease, glioma, aducanumab

## Abstract

This review discusses the current progress in the clinical use of magnetic resonance-guided focused ultrasound (MRgFUS) and other ultrasound platforms to transiently permeabilize the blood-brain barrier (BBB) for drug delivery in neurological disorders and neuro-oncology. Safety trials in humans have followed on from extensive pre-clinical studies, demonstrating a reassuring safety profile and paving the way for numerous translational clinical trials in Alzheimer’s disease, Parkinson’s disease, and primary and metastatic brain tumors. Future directions include improving ultrasound delivery devices, exploring alternative delivery approaches such as nanodroplets, and expanding the application to other neurological conditions.

## 1. Introduction

The blood-brain barrier (BBB) is a semi-permeable barrier between blood vessels and the brain parenchyma, comprising tight junctions between endothelial cells and efflux transporters which actively remove substances from the central nervous system. Ions and small lipid-soluble molecules that are less than 400 Daltons (Da) are often able to pass through the BBB, but larger molecules are unable to gain entry [1]. While essential for maintaining CNS composition and an immune-privileged environment, the BBB also hinders potentially transformative therapies from reaching their intended targets in the brain [2,3].

Numerous strategies for BBB permeabilization are under investigation. Broadly, these strategies can be categorized as transcellular and paracellular [4]. In transcellular approaches, molecules can be made more lipophilic to promote passage across the BBB, or carrier-mediated transport can be enhanced to bypass the BBB altogether [5]. Transcellular approaches can be limited by pharmaceutical agents compatible with these types of manipulation. Paracellular methods involve the disruption of tight junctions, and this can be performed through chemical or physical means. Chemical paracellular mechanisms of BBB permeabilization often rely on vasoactive agents, hyperosmolar compounds (such as mannitol), or antibodies to the claudin family of proteins (integral to the function of tight junctions). Chemical paracellular techniques are promising but often suffer from issues with not being targetable and having off-target effects [4]. 

Paracellular physical mechanisms of BBB permeabilization are gaining traction. In particular, the use of low-intensity ultrasound, combined with the intravenous injection of microbubbles, has emerged as a safe, reproducible, and targeted method for transiently permeabilizing the BBB conformally in a variety of brain structures. Ultrasound-mediated BBB opening has been performed safely across a wide range of pre-clinical models [6], which has led to reassuring safety trials in humans [7]. Figure 1A illustrates three common devices in use today for ultrasound-mediated BBB opening. 

The most common use of ultrasound-mediated BBB permeabilization in human clinical trials is magnetic resonance-guided focused ultrasound (MRgFUS). MRgFUS involves a helmet lined with a phased array of focused ultrasound (FUS) transducers, which is used within an MRI under real-time imaging guidance. Conveniently, the MRI can then also be used to monitor gadolinium extravasation as a marker for successful BBB opening. When performed at high intensities, MRgFUS can create thermo-ablative lesions, which is useful in the treatment of some movement disorders such as tremors, and psychiatric diseases such as obsessive-compulsive disorder, amongst other expanding indications [8,9]. Low intensity ultrasound (for BBB permeabilization) uses approximately 0.01% of the energy required for ablation [10]. When performed without microbubble injection, MRgFUS has also demonstrated utility for the purposes of neuromodulation [11].

**Figure 1 pharmaceutics-16-00719-f001:**
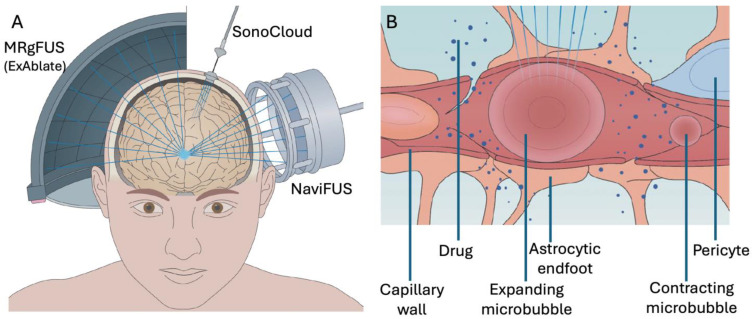
Ultrasound-mediated drug delivery systems and mechanisms. (**A**) Three systems in use are illustrated. The ExAblate system utilizes a helmet lined with a phased array of ultrasound transducers separated from the scalp by cooled degassed water and operated within an MRI. The SonoCloud is an implanted device placed through a burr hole in the skull either at the time of a tumour resection surgery, or with an independently planned procedure. It is powered through a transdermal needle connected to an external power supply for each treatment. NaviFUS is a multi-channel array, like the ExAblate system, but utilizes a smaller surface area and is not performed within an MRI. (**B**) When microbubbles pass through sonicated tissue, they undergo cavitation, causing mechanical forces on the capillary wall, astrocytic endfeet, and pericytes, causing the temporary opening of the blood-brain barrier, allowing larger molecules to pass through into the parenchyma. This figure was adapted with permission from Figures 2 and 3 in Meng et al. (2021) [11].

Another use of ultrasound-mediated BBB permeabilization is through the use of an implantable device. The SonoCloud is implanted in a 3 cm burr hole, either at the time of another planned surgery (i.e., resection of a tumor) or through a separate surgery. The device is powered through a transdermal needle at the time of each treatment. While this method lacks the spatial targeting offered by MRgFUS, it may be advantageous for instances where numerous treatments are required.

Following BBB opening, systemic drug administration can lead to spatially and temporally targeted passage of molecules into the CNS, ranging from small molecules to large monoclonal antibodies or even neural stem cells [12]. This approach is gaining traction with human studies in several conditions, including Alzheimer’s disease (AD), Parkinson’s disease (PD), and primary/secondary brain tumors [13,14]. This mini-review will outline the use of ultrasound-mediated BBB permeabilization in human clinical trials.

## 2. Mechanism of Ultrasonic BBB Opening

Microbubbles are small spheres filled with a high molecular weight gas and lined with a thin microsphere shell, which are then injected peripherally into the bloodstream. Originally developed as a contrast agent for diagnostic cardiac ultrasound, microbubble injection has been repurposed for BBB opening with ultrasound [15]. As the microbubbles pass through a region of ultrasound sonication, as defined conformally with a high degree of accuracy on patient-specific brain MRIs, they undergo oscillations that cause a transient disruption in the BBB (Figure 1B). The extent of BBB disruption is related to acoustic pressure, sonication duration, and the size of microbubbles [16,17]. While these oscillation-related forces open the BBB, if too great, they can lead to hemorrhage. Oscillations induce stretching, acoustic streaming, and shear forces on the vasculature, which can affect the permeability of tight junctions and the function of efflux transporter proteins [18,19]. There are several varieties of microbubbles available on the market, and an ideal agent is one which induces stable cavitation, decreases the nearby production of P-glycoprotein, and results in the formation of caveolae (invaginations in the plasma membrane) [20,21]. A study comparing three commercially available types of microbubbles found all three to perform equivalently in terms of degree and persistence of BBB permeabilization [22]; based on the size and half-life of the microbubble, ultrasound parameters (power and duration) can be varied slightly to optimize MB performance [22].

Early pre-clinical studies were vital in determining a safe range of parameters for FUS. They showed that using lower frequency and peak pressures is consistently safe, creating harmonic bubble oscillations rather than bubble collapse, thereby not damaging blood vessels, neurons, or glia [23]. A mechanical index (MI) was described, calculated as the peak negative pressure (estimated in situ) divided by the square root of the ultrasound frequency; an MI ≤ 0.45 has consistently demonstrated safe outcomes without hemorrhage [24].

BBB permeabilization is a dynamic process, occurring almost immediately following the ultrasound sonication of microbubbles, with confirmation of closure via histology and non-invasive imaging like MRI at approximately 3–24 h [7,25,26]. A T1 MRI with gadolinium infusion is the most commonly used method for demonstrating opening and, later, the restoration of the BBB. While initial human trials sonicated a relatively small volume of tissue (1 cm^3^) in a single treatment, more recently, it has been shown to be safe to sonicate larger volumes (up to 40 cm^3^) with repeated treatments [27] (even larger volumes have been sonicated in non-human primates [28]).

Alternative approaches using ultrasound to enhance drug delivery are also under active investigation. For example, nanodroplets, which have a longer half-life than microbubbles, can be loaded with a therapeutic agent, administered systemically, and when passing through a focal area of sonication, will vaporize, leading to the local delivery of the drug [29]. This technique has been used to deliver phenobarbital to the amygdala for the treatment of agitation in an AD mouse model [30]. Recently, an innovative technique was reported in which rodents were administered a piezoelectric nanogenerator, which embeds into neuronal membranes and, when sonicated with FUS, stimulates tyrosine hydroxylase activity, thus boosting dopamine production in striatal neurons [31]. While the nanodroplet and nanogenerator fields may soon bring revolutionary developments, the reassuring safety and reliability data supporting transcranial and implantable FUS devices for BBB opening has enabled numerous translational human clinical trials to be completed, with many more ongoing.

## 3. Alzheimer’s Disease

AD is a progressive neurodegenerative condition that is the most common cause of dementia. AD etiology is closely tied to the accumulation of toxic extracellular amyloid-beta (Aβ) plaques, intracellular neurofibrillary tangles of tau-protein, neuronal loss (particularly along the circuit of Papez), and dysfunction of the default mode network [32,33,34,35]. Aβ plaques occur at an increased rate throughout the brain of individuals with AD, and they accumulate at sites of vulnerability along nodes of the default mode network (DMN); this Aβ accumulation along the DMN scales with cognitive decline [36,37]. There is substantial interest in utilizing targeted pharmacotherapies for Aβ, such as monoclonal antibodies like aducanumab and lecanemab, to clear Aβ plaques from the CNS [38]. The FDA approved aducanumab on the basis of Aβ clearance alone in 2021 [39], followed by approval of lecanemab in 2023, after a randomized controlled trial suggested both Aβ clearance and modest mitigation of cognitive decline [40]. Despite a half-life of 15–20 days, only 0.01% of Aβ immunoglobins cross the BBB [41]. As most targeted therapies involve large molecules normally filtered by the BBB, there is a strong rationale for delivery using ultrasound (see Table 1) [42].

A 2018 report described the first use of MRgFUS to safely open the BBB in five patients with AD [7]. With the primary goal of demonstrating safety, the BBB was opened in a small volume (approximately 9 × 9 × 9 mm) twice, each 1 month apart [7]. A T1 gadolinium-enhanced MRI demonstrated that BBB opening was transient, closing within 24 h [7]. Other studies then demonstrated the safety of opening the BBB in a larger volume and alternative anatomical targets [43,46]. Recently, MRgFUS-mediated BBB opening was carried out at multiple large-volume nodes of the DMN, such as the bilateral hippocampi and precuneus [27], setting the stage for drug delivery to these and other sites.

Based on pre-clinical studies in which FUS BBB opening resulted in a 5–8 fold increase in CNS penetration of aducanumab [48,49], a recent first-in-human trial was published using MRgFUS to enhance aducanumab delivery in three patients with AD [47]. Six monthly MRgFUS-mediated BBB treatments were paired with increasing doses of intravenous aducanumab infusion. The targeting approach was escalated between patients, sonicating 10 mL in the non-dominant frontal lobe of patient 1 and sonicating 40 mL in the dominant frontal/temporal/hippocampus regions of patient 3. Using contralateral, homologous, untreated brain regions as a control for each patient, marked reductions in Aβ were appreciated on florine-18 florbetaben positron emission tomography (PET) [47]. Patient 3 experienced cognitive deterioration noted at the 30-day follow-up timepoint, the interpretation of which illustrates the difficulty in the AD field of distinguishing expected clinical deterioration from treatment-related adverse effects. This is further challenged by remaining uncertainties, regarding whether reductions in Aβ will translate into improved cognition [50].

## 4. Neuro-Oncology

Improvements in outcomes in neuro-oncology have lagged behind other fields of oncology, due in part to the BBB blocking access of targeted chemotherapeutics to the CNS [51]. Neuro-oncology is one of the most active aspects of research in ultrasound-mediated BBB disruption. Here, we provide a limited overview of the recent studies and future directions in neuro-oncology (see Table 2).

Although the blood-tumor barrier of CNS tumors is altered, larger molecule chemotherapeutics continue to have limited penetration [57,58]. Pre-clinical studies have demonstrated the feasibility, safety, and anti-tumoral effects of FUS-mediated BBB opening when paired with chemotherapies [59]. FUS has been used to enhance the delivery of relatively small molecules, ranging from 150 Da to 1 kDa, such as carmustine (BCNU) [60], etoposide [61], cisplatin [62], irinotecan [63], carboplatin [59], and doxorubicin [64,65]. Even if a substance is able to cross the BBB under normal circumstances (such as BCNU), issues with the half-life and systemic toxicity strengthen the rationale for enhanced targeted delivery. With similar success, larger molecules, ranging from 70 to 150 kDa have been delivered with the help of FUS [64], such as trastuzumab [66], pertuzumab [67], adeno-associated virus [68], IL-12 [69], and a radio-labeled form of bevacizumab [70].

The first-in-human study using MRgFUS, demonstrated the safety of BBB-opening in peri-tumoral tissue in five patients with high-grade gliomas [52]. Participants then underwent surgical resection, and in two patients, pathology specimens were collected of sonicated and non-sonicated tissues. The chemotherapy concentration appeared to be increased in the sonicated area in two patients, however, the overall low drug levels limited the interpretation of this result [52]. This study demonstrated the safety and feasibility of incorporating a BBB-opening procedure into the care of patients with aggressive brain tumors, paving the way for several now-completed and other ongoing trials in this field.

Another study of MRgFUS-mediated BBB opening demonstrated enhanced penetration of fluorescein, which was infused shortly after the FUS procedure, with both visual and biochemical confirmation [55]. A multi-center trial further examining the safety and effect of MRgFUS-mediated BBB opening, combined with maintenance temozolomide therapy in patients with newly diagnosed glioblastoma multiforme, has been concluded with results pending publication (NCT03616860, www.clinicaltrials.gov, accessed 1 March 2024). Accumulating experience with this technology within studies that also incorporate longer follow-ups will help understand whether MRgFUS impacts the progression-free survival or overall survival of high-grade glioma patients.

In a study of four patients with progressive Her2-positive breast cancer metastases, MRgFUS-mediated BBB opening of the tumor and its periphery was paired with the administration of Indium-111-radiolabeled trastuzumab [14]. Single-photon emission computed tomography (SPECT) demonstrated significantly enhanced chemotherapy delivery to the lesion, which was associated with stable or reduced tumor volume during the study [14]. This study also included unique target locations within the posterior fossa, which were targeted safely and effectively [14]. Building on these experiences, a multi-center study to examine the safety of the procedure in patients with diffuse midline glioma is currently underway (NCT05615623). An ongoing multicenter study is looking to compare the MRgFUS-enhanced delivery of pembrolizumab, an immune checkpoint inhibitor, to non-enhanced delivery in patients with non-small cell lung cancer with brain metastases (NCT05317858) [21].

Data from the fluorescein and radiolabeling studies provided important data on the pharmacokinetics of drug delivery using MRgFUS. Unfortunately, the knowledge gaps in the pharmacokinetic profile remain a difficult hurdle in incorporating MRgFUS into standard lines of oncologic treatment. Fluid biomarkers and advanced imaging offer less invasive ways of gathering this type of data. Window of opportunity studies also play an important role in this respect [71].

## 5. Parkinson’s Disease

Parkinson’s disease (PD) is the most common neurodegenerative movement disorder, affecting 1% of those over age 60 [72]. The primary motor symptoms of PD are attributed to dopaminergic neuronal loss in the substantia nigra pars compacta associated with Lewy pathology, leading to a shortage of dopamine in the striatum [73]. The etiology of PD remains an area of investigation, but increasingly, it appears to be an interaction between lifestyle, environment, and genetics [74]. While aging is the primary risk factor for PD [75], other risk factors have been identified, such as exposure to toxins like rotenone and paraquat [76] or gene mutations including, but not limited to, mutations in the *GBA1* [77,78] or *LRRK2* gene [79]. As with AD and neuro-oncology, potential therapies for PD have been limited by the BBB, prompting considerable interest in the FUS-enhanced delivery of agents such as alpha-synuclein targeted therapies or neurotrophic factors to mitigate neuronal loss within the basal ganglia [80].

In pre-clinical models, numerous studies have demonstrated that the BBB can be opened safely in the putamen [81]. When paired with the administration of an alpha-synuclein silencing viral vector or neurotrophic factors, it can attenuate nigrostriatal neuronal loss in MPTP or transgenic mice [82,83]. A recent article describes MRgFUS-mediated BBB opening paired with systemic administration of an adeno-associated virus (AAV) vector, eliciting novel protein expression in the putamen and substantia nigra, highlighting a promising avenue for viral-vector protein expression modulation in patients with PD [28]. However, significant challenges remain with this approach. The cost of systemic delivery of an AAV at sufficient titers to achieve transduction in humans, as well as risks associated with systemic exposure, even with the enhanced BBB opening, will require further investigation. Certainly, advances in AAV serotypes that are neuron-selective and have limited systemic uptake may provide a path forward to allow for more practical combinations of gene therapy technology with MRgFUS [84].

In humans with PD, MRgFUS-mediated BBB opening in the putamen has been shown to be well tolerated with repeat treatments, and there is some early experience with bilateral and repeat treatments (see Table 3) [13,85]. In PD, the putamen is known to be exquisitely sensitive to physical insult [86], and thus far, MRgFUS-mediated BBB opening does not appear to worsen dopaminergic denervation-based radiographic measures, although the existing data are still limited [85], further investigation is needed to understand the thresholds.

The first use of MRgFUS-mediated BBB opening paired with drug delivery in humans with PD was recently described in PD patients with *GBA1* mutations (Gaucher’s disease) [13]. *GBA1* encodes for the enzyme glucocerebroside (GCase), and pre-clinical work suggests that the deficiency or reduced activity of GCase can lead to the accumulation of alpha-synuclein and reversal of this may reduce dopamine neuron cell loss [87,88]. Intravenously administered recombinant GCase is a safe and effective therapy in patients with PD in the context of *GBA1* mutation, but its BBB penetration is poor [89].

Therefore, a study was undertaken to test the safety of combining MRgFUS-mediated BBB opening concurrently with intravenous GCase administration in *GBA1*-related PD [13,90]. In all four patients, a total of three MRgFUS-mediated BBB opening procedures were performed with an increasing dose of GCase administered with each treatment, spaced by two weeks between treatments. A gadolinium-enhanced MRI demonstrated successful BBB opening unilaterally in the putamen without serious adverse effects [13]. Two patients experienced a transient increase in dyskinesia, which was theorized to be related to increased levodopa exposure due to BBB permeabilization [13]. Positron emission tomography demonstrated reduced putamenal metabolism at 1 month, which has been shown to correlate with treatment efficacy in other studies [91]. While this was a small phase 1 trial, it has charted a way forward for targeted delivery of GCase or other molecules (such as neurotrophins or monoclonal antibodies) in the treatment of movement disorders. Future studies will continue to better understand the safety profile of MRgFUS in patients with Parkinson’s disease and the pharmacodynamics of GCase in the brain once delivered via ultrasound.

## 6. Future Directions

In the near future, interest in the use of MRgFUS-mediated BBB opening with drug delivery will continue to increase in AD, PD, neuro-oncology, and numerous other indications. With increased cohort sizes, the biological effect of drugs in the brain being delivered in this fashion will be better understood, allowing for optimized peri-procedural dosing and timing of re-treatment(s). Depending on the indication, randomized controlled trials may allow for better estimates of efficacy. Improvements in the MRgFUS device itself are also expected to yield advancements in the field. Frameless technologies and workflows that require only a minimal haircut are already in early use.

## 7. Conclusions

After extensive pre-clinical development, low-intensity MRgFUS microbubble sonication has emerged as a safe and feasible method of BBB opening in humans. MRgFUS-mediated BBB disruption, paired with systemic drug administration, allows for a drug previously unable to freely bypass the BBB to pass into the targeted region over the ensuing hours. This method has now been applied in human trials in pathologies including AD, Parkinson’s disease, and primary and metastatic cancers. The future of MRgFUS-mediated BBB opening for drug delivery will include a less invasive treatment interface and more seamless workflows to allow for better incorporation with standard treatment paradigms.

## Figures and Tables

**Table 1 pharmaceutics-16-00719-t001:** FUS-mediated BBB-opening in Alzheimer disease.

Study	Condition, Subjects	Device, Treatments, Parameters, Targets	Findings
Lipsman et al., 2018 [7]	AD 6 subjects	ExAblate helmet array + MB injections 2 treatments, 1 month apart 220 KHz, Power = 50% of cavitation threshold—average of 4.5 W Right frontal lobe (DLPFC), 1 cm^3^	MRI-gad confirmed BBB opening, closed at 24 h. No SAEs, 1 patient with transient neuropsychiatric symptoms 2 patients with transient gradient echo changes, nearly resolved by 24 h—possible microhemorrhages No significant change in amyloid at target (PET)
Park et al., 2021 [43]	AD 6 subjects	ExAblate helmet array + MB infusion 2 treatments, 3 months apart 220 KHz, 8–40 W, targeting cavitation of 0.4–0.65 Bilateral frontal lobes, 21 cm^3^	No adverse events MRI-gad confirmed BBB opening in 96% of targeted region 1.6% reduction in PET-measured amyloid in frontal lobes Stable MMSE scores, transient improvement in neuropsychiatric symptoms
Epelbaum et al., 2022 [44]	AD 10 subjects	SonoCloud implantable device + MB injections 7 treatments, every 2 weeks 1 MHz, 0.9–1.03 MPa Implanted overtop the left supramarginal gyrus, explanted after 9 months	1 patient withdrawn due to a thick scalp—unable to activate the implanted device. 1 hemorrhage remote from the implanted device, causing acute delirium. Treatments lasted 4.5 min Possible reduction in amyloid around implant (non-significant)
Meng et al., 2023 [27]	AD 9 subjects	ExAblate helmet array + MB infusion 3 treatments, 2 weeks apart 220 KHz, Power = 50% of cavitation threshold Targeting the DMN: bilateral precuneus and ACC, and bilateral or unilateral hippocampi (started unilateral, advanced to bilateral after safe in first patients); 9 cm^3^	Successful BBB opening and no SAEs. 2 patients had acute confusion, lasting a week in one patient (P9), who was excluded from further procedures. 2 patients with immediate gradient echo changes, resolved the following day Small but significant reduction in PET-measured beta amyloid in sonicated right parahippocampal/inferior temporal region.
Rezai et al., 2022 [45] (longterm data from Rezai et al., 2020 [46])	AD 10 subjects	ExAblate helmet array + MB injections/infusion 3 treatments, 2 weeks apart 220 kHz Initially unilateral hippocampus/EC, 2–5 cm^3^, increased to include frontal and parietal targets (up to 30 cm^3^) in final patients	No SAEs Hippocampal edema in 1 patient, resolved at 72 h. MRI-Gad showed immediate BBB opening in 82% of targeted brain volume, complete closure within 24–48 h. ADAS-Cog/MMSE showed cognition stable at 6 months, mild (expected) decline at 12 months.
Rezai et al., 2024 [47]	AD 3 subjects	ExAblate helmet array + MB infusions 6 monthly treatments paired with aducanumab infusion 2 h prior 220 kHz Patient 1: right frontal lobe (10 mL), patient 2: left frontal/parietal lobe (20 mL), patient 3: left frontal/parietal/temporal lobes & hippocampus (40 mL).	Cognitive worsening in patient 3 a 30 days, not associated with changes in activities of daily living Significant reduction PET-measured amyloid in targeted regions, compared to baseline and untreated contralateral regions

ACC: anterior cingulate cortex; ADAS-Cog: Alzheimer disease assessment scale–cognitive subscale; DLPFC: Dorsolateral prefrontal cortex; DMN: Default mode network; EC: entorhinal cortex; MB: microbubble; MHz: Megahertz; MMSE: Mini-mental state examination; MPa: Megapascal; PET: positron emission tomography; SAE: serious adverse event; W: watts.

**Table 2 pharmaceutics-16-00719-t002:** FUS-mediated BBB-opening in neuro-oncology.

Study	Condition, Subjects	Device, Treatments, Parameters, Targets	Findings
Mainprize et al., 2019 [52]	GBM 5 subjects	ExAblate helmet array + MB injections 1 treatment prior to resection 220 KHz, 50% of cavitation threshold 9 × 9 × 6 mm^3^ at the tumour periphery	MRI-Gad BBB opening in peri-tumour tissue immediately following sonication. Increased chemotherapy concentration in sonicated tumour in 2 patients
Idbaih et al., 2019 [53] (longterm data from Carpentier et al., 2016 [54])	Recurrent GBM 19 subjects	SonoCloud1 implantable device + MB injections, followed by IV carboplatin Mean of 3 treatments, separated by 4 weeks 1 MHz, 0.41–1.15 MPa (dose escalation)	BBB opening occurred in 52/65 sonications (improved with increased acoustic pressure) 2 patients experienced post-sonciation edema, responsive to steroids Authors suggest patients with grade 2/3 BBB opening trended towards longer PFS and OS
Anastasiadis et al., 2021 [55]	Infiltrating glioma (WHO grade 2 and 3) 4 subjects	ExAblate helmet array + MB injection, followed by fluorescein injection and surgical resection. 230 KHz, Power = 50% of cavitation threshold (ranging from a mean of 3–26 W). 0.5 cm^3^ in subjects 1–3, 10 cm^3^ in subject 4.	No SAEs MRI-Gad, and intraoperative visualization/histopathology of fluorescence confirmed increased BBB opening in targeted regions
Meng et al., 2021 [14]	Her2+ breast metastases to brain 4 subjects	ExAblate helmet array + MB infusion 220 KHz, mean power of 13 W Entire tumour volumes targeted, mean volume of 27 cm^3^	Possible to target entire tumour volumes, including lesions in the posterior fossa No SAEs Trastuzumab delivery confirmed with SPECT-imaging Reduction in tumour volume in all 4 patients.
Sonabend et al., 2023 [56]	Newly resected recurrent GBM 17 subjects	SonoCloud9 implantable device + MB injection Up to 6 treatments (median 3) immediately followed with IV paclitaxel administration, dose escalation. Borders of resection cavity, mean volume of 12.6 mL	No SAEs Effective BBB opening, with restoration occurring as early as 1 h. Sonications resulted in transient neurologic deficits (paresthesias, aphasia) related to brain structures adjacent to sonication fields. 2 patients developed reversible encephalopathy at higher paclitaxel dosing

BBB: blood brain barrier; GBM: Glioblastoma multiforme; IV: intravenous; OS: overall survival; PFS: progression-free survival; SPECT: Single-photon emission computed tomography; WHO: world health organization.

**Table 3 pharmaceutics-16-00719-t003:** FUS-mediated BBB-opening in Parkinson’s disease.

Study	Condition, Subjects	Device, Treatments, Parameters, Targets	Findings
Meng et al., 2022 [13]	PD, GBA1 mutation 4 subjects	ExAblate helmet array + MB infusion 3 treatments paired with infusion of GCase, separated by 2 weeks 220 KHz, mean of 6 W Mean target of 3.4 cm^3^, covering 66% of the unilateral putamen	66% of the putamen (unilateral) No SAEs 2 patients developed transient contralateral dyskinesias. 1 transient microhemorrhage detected on T2* and resolved on the following scan Reduction in putaminal hypermetabolism on PET
Pineda-Pardo et al., 2022 [85]	PD 7 subjects	ExAblate helmet array + MB injections 2 treatments, separated by 2–4 weeks. 220 KHz, <15 W Posterior putamen (unilateral in 1st treatment, 3 patients treated bilaterally in 2nd)	No SAEs 2 subjects with transient vasogenic edema at target 5/7 had persistent SWI hypointensities, thought to be microhemorrhages Stable UPDRS scores PET imaging revealed stable dopamine synthesis capacity, and local clearance of amyloid.

GCase: Glucocerebrocidase; PD: Parkinson’s disease; PET: positron emission tomography; SAE: serious adverse events; UPDRS: Unified Parkinson’s Disease Rating Scale; W: Watts.

## Data Availability

Not applicable.

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
