# Peer review of "Current Progress in Magnetic Resonance-Guided Focused Ultrasound to Facilitate Drug Delivery across the Blood-Brain Barrier"

_pharmaceutics, 2024, doi:10.3390/pharmaceutics16060719_

Round 1

Reviewer 1 Report

Comments and Suggestions for Authors

The review is very sparse and fragmentary. The lack of illustrative material has a bad effect on the perception of the material. It is recommended to add diagrams illustrating the mechanism of action of this technology, and provide clinical trial cases in a separate table.

It is necessary to present the advantages and disadvantages of this method of inducing BBB permeability in comparison with other technologies. Since there are not very many clinical trials in this area, the most striking examples of the use of this technology at the preclinical stage and pioneering works should be discussed.

It may be worth describing whether this technology involves the use of any materials and compounds that increase sensitivity to MRgFUS or whether it is a completely non-invasive technology.

Author Response

Reviewer 1:

The review is very sparse and fragmentary. The lack of illustrative material has a bad effect on the perception of the material. It is recommended to add diagrams illustrating the mechanism of action of this technology, and provide clinical trial cases in a separate table.

Thank you for these suggestions. We have added an illustrative figure, and 3 summary tables.

It is necessary to present the advantages and disadvantages of this method of inducing BBB permeability in comparison with other technologies. Since there are not very many clinical trials in this area, the most striking examples of the use of this technology at the preclinical stage and pioneering works should be discussed.

Thank you for this comment. There are now innumerable techniques under investigation for BBB disruption, and it’s not possible to outline each of them. However, your suggestion is well-taken, and we have added the following paragraphs, broadly outlining these techniques, and a few drawbacks and issues which might lead one to adopt using focused ultrasound for drug delivery:

Numerous strategies for BBB-permeabilization are under investigation. Broadly, these strategies can be categorized as transcellular and paracellular [4]. In transcellular approaches, molecules can be made more lipophilic to promote passage across the BBB, or carrier-mediated transport can be enhanced to bypass the BBB altogether [5]. Transcellular approaches can be limited to pharmaceutical agents compatible for these types of manipulation. Paracellular methods involve disruption of tight junctions, and this can be performed through chemical or physical means. Chemical paracellular mechanisms of BBB-permeabilization often rely on vasoactive agents, hyperosmolar compounds (such as mannitol), or antibodies to the claudin family of proteins (integral to the function of tight junctions). Chemical paracellular techniques are promising, but often suffer from issues with not being targetable, and having off-target effects [4].

Paracellular physical mechanisms of BBB-permeabilization are gaining traction. In particular, the use of low intensity ultrasound is combined with intravenous injection of microbubbles and has emerged as a safe, reproducible, and targeted method for transiently permeabilizing the BBB conformally in a variety of brain structures. Ultrasound-mediated BBB-opening can be performed safely across a wide range of preclinical models [6], which led to safety trials in humans, where MRgFUS BBB also demonstrated a reassuring safety profile [7]. Figure 1a illustrates three common devices in use today for ultrasound-mediated BBB-opening.

The most common use of ultrasound-mediated BBB-permeabilization in human clinical trials, is magnetic resonance-guided focused ultrasound (MRgFUS). MRgFUS involves a helmet, lined with a phased-array of ultrasound transducers, which is used within an MRI under real-time imaging guidance, which can also be used to monitor for gadolinium extravasation as a marker for increased BBB permeability. When performed at high-intensities, MRgFUS can create thermo-ablative lesions – useful in the treatment of some movement disorders such as tremor and psychiatric disease including obsessive compulsive disorder amongst other expanding indications [8,9]. When performed at lower-intensities - approximately 0.01 % of the energy used for ablation [10] -  MRgFUS has demonstrated utility for the purposes of BBB permeabilization or neuromodulation [11].

Another use of ultrasound-mediated BBB-permeabilization is through the use of an implantable device. The SonoCloud is implanted in a 3cm burr hole, either at the time of another planned surgery (i.e. resection of a tumour), or through a separate surgery. The device is powered through a transdermal needle at the time of each treatment. While this method lacks the spatial-targeting offered with MRgFUS, it may be advantageous for instances where numerous treatments are required.

It may be worth describing whether this technology involves the use of any materials and compounds that increase sensitivity to MRgFUS or whether it is a completely non-invasive technology.

Thank you for this comment. We would refrain from using the term ‘non-invasive’ with any treatment, but rather to think of invasiveness along a spectrum.

Reviewer 2 Report

Comments and Suggestions for Authors

Dear Authors,

I hope this message finds you well. I have completed my review of the manuscript titled "  Current progress in magnetic resonance-guided focused ultrasound to facilitate drug delivery across the blood brain barrie" After careful consideration, I am pleased to recommend major revisions for the manuscript's publication in Pharmaceuticals.

Comments:

*Please check the title.Current progress in magnetic resonance-guided focused ultrasound to facilitate drug delivery across the blood brain barrie

* Kindly revise the manuscript for grammar errors in the text, as some sentences appear complex.

Original: While essential for maintaining CNS composition and maintaining an immune-privileged environment, the BBB also hinders potentially transformative therapies from reaching their intended targets in the brain[2,3]. 

Recommended: While essential for maintaining CNS composition and an immune-privileged environment, the BBB also hinders potentially transformative therapies from reaching their intended targets in the brain [2,3].

Original: Low intensity FUS is combined with intravenous injection of microbubbles and has emerged as a safe, reproducible, and targeted method for transiently permeabilizing the BBB conformally in a variety of brain structures. 

Recommended: Low-intensity FUS, combined with intravenous injection of microbubbles, has emerged as a safe, reproducible, and targeted method for transiently permeabilizing the BBB conformally in a variety of brain structures.

Original: The addition of MRI as in MRgFUS can be helpful again to delineate the target and monitor for gadolinium extravasation as a marker for increased BBB permeability. 

Recommended: The addition of MRI, as in MRgFUS, can be helpful again to delineate the target and monitor for gadolinium extravasation as a marker for increased BBB permeability.

Etc.

*I would recommend going into more detail in the section that discusses the various types of microbubbles and asking the author to give concrete instances to highlight the qualities of the perfect microbubble agent.

* Please check all abbreviations. (for example: Alzheimer disease (AD))

* This sentence should be reconstructed meaningfully. sonicating just 10mL?

The targeting-approach was escalated between patients, sonicating just 10mL in the nondominant frontal lobe of the patient 1, and sonicating 40mL in the dominant frontal/temporal/hippocampus regions of patient 3.

Please check the referencing status as per the journal policy.

For example: A multi-center trial further examining the safety and effect of MRgFUS BBB opening combined with maintenance temozolomide therapy in patients with newly diagnosed glioblastoma multiforme is concluded with results pending publication (NCT03616860, www.clinicaltrials.gov).

*I recommend developing a table that integrates data from the literature to showcase the latest developments in magnetic resonance-guided focused ultrasound (MRgFUS) for improving drug delivery across the blood-brain barrier. This table should encompass information regarding drug release mechanism, efficacy, and biocompatibility.

I am confident that these major revisions will elevate the manuscript's quality and clarity. Your attention to these suggestions is crucial for ensuring the manuscript's successful publication in Pharmaceuticals.

Best regards,

Comments on the Quality of English Language

Dear Editor,

I hope this message finds you well. I have completed my review of the manuscript titled "  Current progress in magnetic resonance-guided focused ultrasound to facilitate drug delivery across the blood brain barrie" After careful consideration, I am pleased to recommend major revisions for the manuscript's publication in Pharmaceuticals.

Comments:

*Please check the title.Current progress in magnetic resonance-guided focused ultrasound to facilitate drug delivery across the blood brain barrie

* Kindly revise the manuscript for grammar errors in the text, as some sentences appear complex.

Original: While essential for maintaining CNS composition and maintaining an immune-privileged environment, the BBB also hinders potentially transformative therapies from reaching their intended targets in the brain[2,3]. 

Recommended: While essential for maintaining CNS composition and an immune-privileged environment, the BBB also hinders potentially transformative therapies from reaching their intended targets in the brain [2,3].

Original: Low intensity FUS is combined with intravenous injection of microbubbles and has emerged as a safe, reproducible, and targeted method for transiently permeabilizing the BBB conformally in a variety of brain structures. 

Recommended: Low-intensity FUS, combined with intravenous injection of microbubbles, has emerged as a safe, reproducible, and targeted method for transiently permeabilizing the BBB conformally in a variety of brain structures.

Original: The addition of MRI as in MRgFUS can be helpful again to delineate the target and monitor for gadolinium extravasation as a marker for increased BBB permeability. 

Recommended: The addition of MRI, as in MRgFUS, can be helpful again to delineate the target and monitor for gadolinium extravasation as a marker for increased BBB permeability.

Etc.

*I would recommend going into more detail in the section that discusses the various types of microbubbles and asking the author to give concrete instances to highlight the qualities of the perfect microbubble agent.

* Please check all abbreviations. (for example: Alzheimer disease (AD))

* This sentence should be reconstructed meaningfully. sonicating just 10mL?

The targeting-approach was escalated between patients, sonicating just 10mL in the nondominant frontal lobe of the patient 1, and sonicating 40mL in the dominant frontal/temporal/hippocampus regions of patient 3.

Please check the referencing status as per the journal policy.

For example: A multi-center trial further examining the safety and effect of MRgFUS BBB opening combined with maintenance temozolomide therapy in patients with newly diagnosed glioblastoma multiforme is concluded with results pending publication (NCT03616860, www.clinicaltrials.gov).

*I recommend developing a table that integrates data from the literature to showcase the latest developments in magnetic resonance-guided focused ultrasound (MRgFUS) for improving drug delivery across the blood-brain barrier. This table should encompass information regarding drug release mechanism, efficacy, and biocompatibility.

I am confident that these major revisions will elevate the manuscript's quality and clarity. Your attention to these suggestions is crucial for ensuring the manuscript's successful publication in Pharmaceuticals.

Best regards,

Author Response

Reviewer 2:

Comments:

*Please check the title. Current progress in magnetic resonance-guided focused ultrasound to facilitate drug delivery across the blood brain barrie

Thank you for noticing this. On our end I don’t see the “r” missing from the title, but we will ensure this is not missed in the final publication.

* Kindly revise the manuscript for grammar errors in the text, as some sentences appear complex.

Original: While essential for maintaining CNS composition and maintaining an immune-privileged environment, the BBB also hinders potentially transformative therapies from reaching their intended targets in the brain[2,3]. 

Recommended: While essential for maintaining CNS composition and an immune-privileged environment, the BBB also hinders potentially transformative therapies from reaching their intended targets in the brain [2,3].

Thank you for noticing this – we have adopted your exact suggestion.

Original: Low intensity FUS is combined with intravenous injection of microbubbles and has emerged as a safe, reproducible, and targeted method for transiently permeabilizing the BBB conformally in a variety of brain structures. 

Recommended: Low-intensity FUS, combined with intravenous injection of microbubbles, has emerged as a safe, reproducible, and targeted method for transiently permeabilizing the BBB conformally in a variety of brain structures.

Thank you, we have adopted your suggested edit!

Original: The addition of MRI as in MRgFUS can be helpful again to delineate the target and monitor for gadolinium extravasation as a marker for increased BBB permeability. 

Recommended: The addition of MRI, as in MRgFUS, can be helpful again to delineate the target and monitor for gadolinium extravasation as a marker for increased BBB permeability.

Thank you! As part of other edits, we have truncated and relocated this sentence.

Etc.

*I would recommend going into more detail in the section that discusses the various types of microbubbles and asking the author to give concrete instances to highlight the qualities of the perfect microbubble agent.

Thank you for this excellent point. It would seem that the three commercially-available (and commonly used) microbubbles performed quite similarly for BBB-opening, and we’ve added the following sentence on this topic:

A study comparing 3 commercially available types of microbubbles, found all three to perform equivalently in terms of degree and persistence of BBB-permeabilization [22]; based on the size and half-life of the microbubble, ultrasound parameters (power and duration) can be varied slightly to optimize MB performance [22].

* Please check all abbreviations. (for example: Alzheimer disease (AD))

Thank you for pointing this out – we’ve correct the us of the AD abbreviation, and other abbreviations.

* This sentence should be reconstructed meaningfully. sonicating just 10mL?

Thank you for this observation – we’ve removed the word “just” and it now reads much better.

Please check the referencing status as per the journal policy.

We are using the MDPI-style, which I believe is in line with the journal policy.

For example: A multi-center trial further examining the safety and effect of MRgFUS BBB opening combined with maintenance temozolomide therapy in patients with newly diagnosed glioblastoma multiforme is concluded with results pending publication (NCT03616860, www.clinicaltrials.gov).

Here, there is no published work to reference, so we are referring to this clinicaltrials.gov entry.

*I recommend developing a table that integrates data from the literature to showcase the latest developments in magnetic resonance-guided focused ultrasound (MRgFUS) for improving drug delivery across the blood-brain barrier. This table should encompass information regarding drug release mechanism, efficacy, and biocompatibility.

We have now included 3 tables, summarizing the ultrasound parameters, targets, and clinical data described throughout this review.

I am confident that these major revisions will elevate the manuscript's quality and clarity. Your attention to these suggestions is crucial for ensuring the manuscript's successful publication in Pharmaceuticals.

Thank you for these kind words! And thanks again for these helpful, constructive comments.

Reviewer 3 Report

Comments and Suggestions for Authors

Dr. Meng et al. have summarized a review article focusing on the magnetic resonance-guided focused ultrasound (MRgFUS) as a drug delivery tool for across the BBB. Recently, the microbubble is favorable for one of the BBB opening methods and MRgFUS may be useful for facilitating its technology. Therefore, although it considers that its technology is still immature, summarizing the related technologies will be useful to researchers and will be meaningful for the development of drug delivery to the central nervous system. It would appreciate if authors could address the concerns below.

In line 44 to 46th, authors described that small lipid-soluble molecules with less than 400 Daltons can pass freely through the BBB. There are many lipid-soluble molecules with less than 400 Da that cannot pass through the BBB membrane. This description may confuse readers. It is considered that more detailed descriptions including the molecular mechanism of BBB penetration should be discussed.

   In line 89th, it seems to be a simple mistake that the word “neurons” overlap. please confirm it.

In line 95th, authors described that microbubble could open the BBB over 3 to 24 hours. Although it is maybe temporary, does the fact that the BBB membrane remains open for a relatively long time (3-24 hours) mean that all substances can freely uptake to and clear from the brain during this opening time? If all substances other than the target drug can be transferred to the brain, can it be concluded safe? In addition, this technique appears to have the additional risks if the procedures repeated. Is it safe sufficiently in such a situation?

   In line 158th, authors have described the molecules which are less than 400 Da can freely transport through the BBB, in line 45th. Why should it consider this technique for the small molecules that can be translocated freely? The description on the drugs with 150-400 Da seems to be consistent the description that the small drugs can be freely taken into the brain.

   In line 207 to 219th, authors described on the AAV. AAV is a viral vector that originally migrates into the nerves. How is the synergistic effect between AAV and BBB opening? Also, is AAV selectively taken up in the BBB opening region?

   Throughout, reviewer considered the manuscript would be better if more specific related technologies including types of materials, bubble composition, and MRgFUS methods were listed. Please add the table such as a list of technologies obtained from previous reports.

Author Response

Reviewer 3:

In line 44 to 46th, authors described that small lipid-soluble molecules with less than 400 Daltons can pass freely through the BBB. There are many lipid-soluble molecules with less than 400 Da that cannot pass through the BBB membrane. This description may confuse readers. It is considered that more detailed descriptions including the molecular mechanism of BBB penetration should be discussed.

Thank you, this is an important point of clarification. We do not want to go into much detail on the specifics of BBB physiology, so have revised the sentence in question to read:

Ions and small lipid-soluble molecules which are less than 400 Daltons (Da) are often able to pass through the BBB, but larger molecules are unable to gain entry.

In line 89th, it seems to be a simple mistake that the word “neurons” overlap. please confirm it.

Yes thank you for noticing! The 2nd ‘neurons’ was changed to ‘glia’.

In line 95th, authors described that microbubble could open the BBB over 3 to 24 hours. Although it is maybe temporary, does the fact that the BBB membrane remains open for a relatively long time (3-24 hours) mean that all substances can freely uptake to and clear from the brain during this opening time? If all substances other than the target drug can be transferred to the brain, can it be concluded safe? In addition, this technique appears to have the additional risks if the procedures repeated. Is it safe sufficiently in such a situation?

While the BBB opening is thought to close within hours of the FUS exposure, the range provided (3-24hrs) reflects what is measured/reported in published literature. In clinical trials, the first time follow-up MRI to check for BBB closure is typically at 24 hours. It is possible that blood borne substances (e.g. albumin) can enter the brain, however no clinical adverse events through examinations and neuroimaging have yet been reported in relation to this. Nevertheless, the overall safety profile of repeated FUS exposures continues to be characterized. We have changed the sentence in question to be:

BBB permeabilization is a dynamic process, occurring almost immediately following FUS sonication of microbubbles, with confirmation of closure via histology and non-invasive imaging like MRI at approximately 3-24 hours

   In line 158th, authors have described the molecules which are less than 400 Da can freely transport through the BBB, in line 45th. Why should it consider this technique for the small molecules that can be translocated freely? The description on the drugs with 150-400 Da seems to be consistent the description that the small drugs can be freely taken into the brain.

Thank you for pointing out this discrepancy. The word ‘freely’ was removed, because this is not correct – the BBB may still reduce the amount of small molecules, like BCNU (carmustine) from crossing. The rationale for using ultrasound-mediated BBB-permeabilization for small molecules is to ‘enhance’ delivery, and also to allow lower systemic doses (thus reducing therapy-limiting side effects). We have also added the following sentence:

Even if a substance is able to cross the BBB under normal circumstances (such as BCNU), issues with halflife and systemic toxicity strengthen the rationale for enhanced targeted delivery.  

   In line 207 to 219th, authors described on the AAV. AAV is a viral vector that originally migrates into the nerves. How is the synergistic effect between AAV and BBB opening? Also, is AAV selectively taken up in the BBB opening region?

BBB-opening allows local targeted-uptake of the AAV vector into the brain parenchyma, when the virus in then able to infect neurons and insert genetic material. To clarify this, we have reworded the sentence to:

A recent article describes MRgFUS-mediated BBB opening paired with systemic administration of an adeno-associated virus (AAV) vector, eliciting novel protein expression in the putamen and substantia nigra, highlighting a promising avenue for viral-vector protein expression modulation in patients with PD

   Throughout, reviewer considered the manuscript would be better if more specific related technologies including types of materials, bubble composition, and MRgFUS methods were listed. Please add the table such as a list of technologies obtained from previous reports.

Thank you for this excellent suggestion. We have now created tables 1-3 to address this.

Round 2

Reviewer 1 Report

Comments and Suggestions for Authors

Please, provide the manuscript in readable form, not in PDF with comments and remarks

Reviewer 2 Report

Comments and Suggestions for Authors

Dear Editor,

 I am writing to formally recommend the acceptance of the revised manuscript titled " Current progress in magnetic resonance-guided focused ultrasound to facilitate drug delivery across the blood brain barrier" following its revision by the authors in response to the initial review process. The revisions made have significantly improved the quality and clarity of the manuscript, and I believe the study now makes a valuable contribution to the field. Thank you for considering my recommendation. I trust that the publication of this manuscript will enrich the journal and contribute to advancing the scientific discourse in the relevant field.

Best